# Identification of CD73 as a Novel Biomarker Encompassing the Tumor Microenvironment, Prognosis, and Therapeutic Responses in Various Cancers

**DOI:** 10.3390/cancers14225663

**Published:** 2022-11-17

**Authors:** Kun Tang, Jingwei Zhang, Hui Cao, Gelei Xiao, Zeyu Wang, Xun Zhang, Nan Zhang, Wantao Wu, Hao Zhang, Qianrong Wang, Huilan Xu, Quan Cheng

**Affiliations:** 1Department of Social Medicine and Health Management, Xiangya School of Public Health, Central South University, Changsha 410008, China; 2Department of Discipline Construction, Xiangya Hospital, Central South University, Changsha 410008, China; 3Department of Neurosurgery, Xiangya Hospital, Central South University, Changsha 410008, China; 4National Clinical Research Center for Geriatric Disorders, Changsha 410008, China; 5Brain Hospital of Hunan Province, The Second People’s Hospital of Hunan Province, Changsha 410007, China; 6The School of Clinical Medicine, Hunan University of Chinese Medicine, Changsha 410007, China; 7One-Third Lab, College of Bioinformatics Science and Technology, Harbin Medical University, Harbin 150086, China; 8Department of Oncology, Xiangya Hospital, Central South University, Changsha 410008, China; 9Department of Neurosurgery, The Second Affiliated Hospital, Chongqing Medical University, Chongqing 400010, China; 10Key Laboratory of Diabetes Immunology, National Clinical Research Center for Metabolic Diseases, Central South University, Ministry of Education, Changsha 410011, China; 11Department of Metabolism and Endocrinology, The Second Xiangya Hospital, Central South University, Changsha 410011, China; 12Clinical Diagnosis and Therapy Center for Glioma of Xiangya Hospital, Central South University, Changsha 410008, China; 13Department of Clinical Pharmacology, Xiangya Hospital, Central South University, Changsha 410008, China

**Keywords:** CD73, cancer, immunotherapy, macrophages, T cells

## Abstract

**Simple Summary:**

Immunotherapy targeting immune checkpoints and stromal cells in the tumor microenvironment is currently one of the most promising directions for tumor therapy. Ongoing studies suggest that CD73 plays an important role in the tumor immune process in certain tumors, however, the exact mechanism is unknown. We aim to fully reveal the prognostic value of CD73 in pan-cancer and its role in tumor immunity through large-scale single-cell and bulk sequencing analysis. We found that high CD73 expression was significantly associated with poor prognosis in many tumors. It is also strongly associated with immune scores, stromal cell infiltration, and immune-related pathways. CD73 can regulate the biological behavior of immune cells in the tumor microenvironment, especially macrophages and T cells. Immunotherapy targeting CD73 has obvious effects, and CD73 may shine as a new immune checkpoint in future tumor immunotherapy.

**Abstract:**

CD73 is essential in promoting tumor growth by prohibiting anti-tumor immunity in many cancer types. While the mechanism remains largely unknown, our paper comprehensively confirmed the onco-immunological characteristics of CD73 in the tumor microenvironment (TME) of pan-cancer. This paper explored the expression pattern, mutational profile, prognostic value, tumor immune infiltration, and response to immunotherapy of CD73 in a continuous cohort of cancers through various computational tools. The co-expression of CD73 on cancer cells, immune cells, and stromal cells in the TME was also detected. Especially, we examined the correlation between CD73 and CD8^+^ (a marker of T cell), CD68^+^ (a marker of macrophage), and CD163^+^ (a marker of M2 macrophage) cells using multiplex immunofluorescence staining of tissue microarrays. CD73 expression is significantly associated with a patient’s prognosis and could be a promising predictor of these cancers. High CD73 levels are strongly linked to immune infiltrations, neoantigens, and immune checkpoint expression in the TME. In particular, enrichment signaling pathway analysis demonstrated that CD73 was obviously related to activation pathways of immune cells, including T cells, macrophages, and cancer-associated fibroblasts (CAFs). Meanwhile, single-cell sequencing algorithms found that CD73 is predominantly co-expressed on cancer cells, CAFs, M2 macrophages, and T cells in several cancers. In addition, we explored the cellular communication among 14 cell types in glioblastoma (GBM) based on CD73 expression. Based on the expression of CD73 as well as macrophage and T cell markers, we predicted the methylation and enrichment pathways of these markers in pan-cancer. Furthermore, a lot of therapeutic molecules sensitive to these markers were predicted. Finally, potential anticancer inhibitors, immunotherapies, and gene therapy responses targeting CD73 were identified from a series of immunotherapy cohorts. CD73 is closely linked to clinical prognosis and immune infiltration in many cancers. Targeting CD73-dependent signaling pathways may be a promising therapeutic strategy for future tumor immunotherapy.

## 1. Introduction

Recently, epidemiological studies have shown that cancer has become the leading cause of premature death in at least 50 countries [1]. Worldwide, more than 19 million newly diagnosed cancer cases and over 10 million cancer-related deaths occurred last year [1]. However, oncology treatment remains unsatisfactory despite continuous advances in diagnostic techniques and treatment modalities. In particular, immune evasion mechanisms in the tumor microenvironment (such as induction of anti-apoptotic factors, down-regulation of tumor antigen expression, and secretion of immunosuppressive molecules) can prevent malignant cells from being captured by the inner immune system. TME is a dynamic and complex network consisting of tumor cells, stromal cells, infiltrating immune cells, cellular matrix, and various molecules [2,3]. These infiltrating immune cells in TME, including natural killer cells (NK), tumor-associated macrophages (TAMs), T and B lymphocytes, CAFs, neutrophils, and dendritic cells (DCs), are fundamental in determining the realistic characteristics of the tumor and are tightly associated with the clinical outcome of a patient with cancers [4,5]. Increasing evidence demonstrated that the antibodies or vaccines targeting inhibitory molecules secreted by immune cells might provide promising directions for the development of next-generation immunotherapies [6,7,8]. For example, blocking the cytotoxic T-lymphocyte-associated antigen 4 (CTLA-4) and programmed death 1 (PD-1) signaling pathways has remarkably improved the effectiveness of immunotherapy against many cancer types [9,10,11].

The cluster of differentiation 73 (CD73), also named 5′-nucleotidase Ecto (NT5E), is a surface enzyme that consists of two identical subunits bound by a glycosylphosphatidylinositol linkage to the external membrane [12]. Adenosine regulates anti-inflammatory signaling pathways by linking P1 receptors to infiltrated immune cells in the TME [13]. Evidence suggests that CD73 plays an essential role in balancing inflammation and immune suppression by converting AMP to adenosine. Importantly, studies have shown that CD73 can modulate tumorigenesis, proliferation, migration, and immune escape, and may be a novel immune regulator in tumor immunotherapy. Furthermore, upregulated CD73 protein levels in tumor tissues correlate with adverse clinical prognosis in several tumor types, such as chronic lymphocytic leukemia [14], triple-negative breast cancer [15], urothelial bladder cancer [16], and gliomas [17], which further highlights the vital role of CD73 in tumor development. Currently, antibody therapies to neutralize CD73, either alone or in combination with other small molecular antagonists, are being tested in clinical trials (ClinicalTrials.gov Identifiers: NCT04797468, NCT04148937, NCT03616886) [18]. However, the core mechanisms of CD73 involvement in tumorigenesis and tumor immunity remain largely undefined. In addition, the interaction network between CD73 and infiltrated cells in the microenvironments of these tumors has not been fully elaborated.

Therefore, we propose that CD73 may affect tumorigenesis and progression by influencing the biological functions of immune cells in the tumor microenvironment. To verify our hypothesis, we identified the prognostic value of CD73 in pan-cancer using a variety of algorithms combined with the data from the GEO, GTEx, and TCGA databases in the current research. Meanwhile, we showed the mutational characteristics of CD73 in pan-cancer using several online datasets. In addition, the role of CD73 in tumor immunity and related signaling pathways was investigated, which may provide new insights for tumor immunotherapy.

## 2. Materials and Methods

### 2.1. Data Collecting and Organizing

Pan-cancer data were downloaded from the TCGA dataset [19]. The normal control data were downloaded from the GETX dataset [20]. Normalize the data based on our previous study [21]. Cell lines data were downloaded from the CCLE (GSE36139) and HPA (PRJEB4337) datasets [22,23]. Transcriptomic data from tumor types were analyzed without batch effects. In addition, the tumor types were analyzed independently and the results are shown in one graph. Therefore, the heterogeneity of the tumor types does not affect the results. Single-cell sequencing data was collected as we did before [21].

### 2.2. Identification of CD73-Related Features

Mutations in the CD73 gene were observed on the CBIOPORTAL website [24]. We used the Kaplan–Meier (KM) algorithm to display the overall survival (OS) and disease-specific survival (DSS) [25]. The TIMER 2.0 algorithm was applied to study the infiltration of immune cells in the tumor microenvironment of these cancers [26]. The calculation of the three scores (immune, stromal, and estimate) was performed by the ESTIMATE algorithm. We also showed the association of CD73 levels and the number of these immune cells with these scores. The Sangerbox website (http://past20.sangerbox.com/, 7 July 2021) was used to test the relationship between CD73 levels and immune checkpoints, DNA MMR markers, TMB and MSI.

To detect the role of CD73 in tumor immunity, gene set variation analysis (GSVA) [27] and the Sangerbox [28] were applied to explore the relevant pathways of CD73 based on the Kyoto Encyclopedia of Genes and Genomes (KEGG) [29] and Hallmark [30] gene sets. The correlation between cellular markers (macrophages and T cells) and CD73 was predicted by the Gene Set Cancer Analysis (GSCA) website [27]. We predicted the immunotherapeutic value of CD73 from the TIDE [31] and TISMO [32] websites. As previously mentioned, potential inhibitors targeting CD73 in cancer were predicted [33].

### 2.3. Single-Cell Sequencing Analysis

Data integration of BRCA (GSE75688 and GSE118389) [34,35] was performed using the R package Seurat (anchoring function). Single-cell sequencing datasets for STAD were taken from the GEO database (GSE183904) [36]. Quality control of mitochondrial and ERCC genes was performed using the R package Seurat [37]. Principal component analysis (PCA) and cell clustering were performed as previously [33]. R packages (infercnv and copycat) were conducted to identify tumor cells. The annotation of non-tumor cells (immune and stromal cells) was based on specific markers. The specific markers of non-tumor cells are listed as follows: fibroblast (COL3A1), M2 macrophage (CD163,CD68,MRC1), B cell (CD79A), plasma cell (JCHAIN), T cell (CD3D, CD8A, CD4), NKs (GNLY, NKG7, EOMES, KIR2DL3, GZMA), astrocyte (ALDH1L1, SLC1A3, SLC1A2, GFAP), immune cell (PTPRC), Tregs (FOXP3, IL2RA), DC (CD80, CD86, CD40), FDCs (CR2, FCER2, CR1), plasma cell (SDC1, TNFRSF17), neutrophils (CEACAM8), ILC1 (TBX21, IKZF3, CXCR3), ILC2 (GATA3, MAF, PTGDR2, HPGDS), ILC3 (RORC, IL23R, IL1R1, KIT). Dimensionality reduction for visualization was performed using the UMAP function. Vlnplot, Dimplot, and Featureplot were used to visualize CD73 expression. When collecting samples for RNA bulk sequencing or single-cell sequencing analysis, tumor samples are usually surrounded by some paraneoplastic tissues consisting of infiltrated cells. Thus, immune and stromal cells can be counted as a reflection of the tumor microenvironment during the data analysis. The R package “Monocle” was used to perform ingle-cell pseudo-time trajectory analysis. Cells in the same segment of the trajectory are considered to have the same “state”. The cell–cell interaction was analyzed by the R package “CellChat” [38].

### 2.4. Multiple Fluorescent Staining

Multiple fluorescent staining was carried out according to the previous procedure [39,40]. For immunofluorescence staining, the primary antibodies were CD73 (Rabbit, 1:200, Proteintech, Wuhan, China), CD68 (Rabbit, 1:3000, Servicebio, Wuhan, China), CD163 (Rabbit, 1:3000, Proteintech, Wuhan, China), CD8 (Mouse, 1:3000, Proteintech, Wuhan, China). The primary antibodies are applied, followed by incubation of secondary antibody (GB23301, GB23303, Servicebio, Wuhan, China). DAPI was sequentially applied after incubation with the human antigens. We obtained the multispectral images after scanning slides using a Pannoramic Scanner (3D HISTECH, Budapest, Öv u. 3., Hungary). We performed the preliminary experiment to explore the optimal concentration of antibodies for the multiplex immunofluorescence staining. A standard site of the tissue array was used as a control to determine the optimal signal intensity for a high resolution. CD8 was selected as a T cell surface marker; CD68 was selected as a macrophage surface marker; CD163 was selected as an M2 macrophage surface marker. Positively stained cells were analyzed using Caseviewer (CV 2.3, CV 2.0) image analysis software (Budapest, Öv u. 3., Hungary). Cells with distinct nuclei stained with DAPI and a clear surrounding area stained with CD73 were analyzed to see the expression of CD73. The tissue microarray was obtained from the Outdo Biotech company (HOrg-C110PT-01, Shanghai, China) and the ethics were approved.

### 2.5. Statistical Analysis

Optimal cut-off values for CD73 were calculated by the R package survminer and patients with different survival outcomes were grouped. Survival analysis was performed by univariate Cox regression analysis for OS and DSS in the high CD73 group and low CD73 group in pan-cancer. Two-tailed *t*-test or one-way ANOVA was applied to identify significant differences between groups (normally distributed variables), respectively. Wilcoxon test or the Kruskal–Wallis test was applied to identify significant differences between groups (variables not normally distributed), respectively. All tests were two-way and *p* < 0.05 was considered statistically significant.

## 3. Results

### 3.1. CD73 Expression in Tumor Tissues, Counterparts and Cell Lines

To systematically clarify CD73 expression in the normal tissues and cancers, we first observed CD73 levels based on four public databases-GETx, CCLE, HPA, and TCGA. In the GETx dataset, CD73 was expressed to varying degrees in 31 human tissues, such as blood vessels, cervix uteri, vagina, nerve, uterus, and ovary (Figure 1A). The expression spectrum of CD73 in tumor cell lines demonstrated that CD73 was significantly elevated in almost 21 tumor cell lines except hematopoietic and lymphoid cell lines based on the CCLE dataset (Figure 1B). In addition, we explored the RNA expression of CD73 in 48 tumor cell lines from the HPA dataset (Figure 1C). The top three cell lines with high CD73 expression were U-87 MG, TIME, and U-251 MG, derived from human malignant glioblastoma multiforme and telomerase-immortalized human microvascular endothelium cell lines, respectively. Meanwhile, we displayed the expression landscape of CD73 using the R language after combining the latest data from TCGA and GETx databases (Figure 1D). We found that CD73 was expressed in cancers higher than normal controls, including GBM, LGG, PAAD, KIRP, STAD, KIRC, LAML, ESCA, THYM, LIHC, COAD, PCPG, DLBC, HNSC, READ, and LUAD. In contrast, CD73 levels were upregulated in normal controls than cancers, including OV, CESC, SKCM, TGCT, CHOL, PRAD, KICH, UCEC, BRCA, UCS, BLCA, and LUSC.

### 3.2. Mutational Aspects and Prognostic Role of CD73

Then, we demonstrated the mutation of CD73 in these cancers from the cBioportal website (Appendix A). The level of mutations in DLBC and PRAD was high, and the frequency of deep deletions in CD73 exceeded 4% (Appendix A). We also measured CD93 mRNA expression, mutation type, structural variant, and copy number in these tumors (Appendix A). Seventy-four mutation sites (including 64 missenses, 8 truncating, 1 fusion, and 1 inflame) were detected between 0 and 574 amino acids (Appendix A). Clinical and RNA sequencing data were outputted from public websites to further identify the prognostic role of CD73. Our data indicated that CD73 can accurately predict good or bad OS (Figure 2A) and DSS (Figure 2B) in many tumors. Overexpressed CD73 was related to decreased OS in ACC, BRCA, CESC, HNSC, KIRP, LGG, LIHC, LUAD, LUSC, MESO, STAD, UVM, KICH, PAAD, and TGCT. However, in KIRC, PCPG, PRAD, SARC, and SKCM, upregulated CD73 was related to prolonged OS Figure 2C; (*p* < 0.05). Elevated CD73 was associated with decreased DSS in BRCA, CESC, COAD, ESCA, GBM, HNSC, KIRP, KICH, LGG, LIHC, LUAD, LUSC, MESO, PAAD, STAD, TGCT, OV, and UVM, and with prolonged DSS in CHOL, KIRC, PCPG, SKCM and SARC (Appendix A; *p* < 0.05).

### 3.3. Immune Characteristics of CD73 in the TME

Next, the immune characteristics of CD73 in these cancers were identified. The relationship between CD73 expression and the stromal scores (Appendix A), immune scores (Appendix A), and estimate scores (Appendix A) were calculated in these tumors. Our results showed that CD73 expression is strongly linked to these scores in most cancers. In particular, BLCA, BRCA, KIRC, HNSC, LUAD, MESO, PRAD, SARC, SKCM, TGCT, THCA, THYM, UCEC LUSC, and OV were the cancers most positively linked to stromal scores (*p* < 0.0001); BLCA, PRAD, LUAD, LUSC, OV, SARC, THCA, and BRCA were the cancers most positively linked to immune scores (*p* < 0.0001); BLCA, BRCA, HNSC, LUAD, LUSC, PRAD, SARC, SKCM, THCA, THYM, OV, and PRAD were the cancers most positively linked to estimate scores (*p* < 0.0001). Moreover, the role of CD73 expression in the immune infiltration of these cancer microenvironments was studied. Our data showed that the top three cancers were BLCA, BRCA, and COAD, where CD73 expression in the tumor microenvironment is associated with infiltration of immune cells, including B cell, CD4 and CD8 T cell, dendritic cell, macrophage, and neutrophil (Appendix A; *p* < 0.05). These results demonstrate the important role of CD73 in mediating immune cell migration in these cancers, especially in macrophages and T cells. Neoantigens are mutated genes of tumor cells, which were associated with tumor immunotherapy [41,42]. Next, the relationship between CD73 and neoantigen expression in these cancers was explored (Appendix A). The results suggested that CD73 levels were significantly linked to neoantigen levels of KIRC, HNSC, and CESC (*p* < 0.05).

### 3.4. CD73 Correlated with Checkpoints, MMR Markers, TMB, and MSI

To further elaborate on the potential value of CD73 in immunotherapy, we compared the relationship between CD73 and classical immune checkpoints in various cancers (Figure 3A). The immune checkpoints most linked to CD73 were NRP1, CD276, PDCD1LG2 (PD-L2), CD274 (PD-L1), and CD44. We also analyzed the correlation between CD73 and five DNA mismatch repair (MMR) markers, including MLH1, MSH2, MSH6, PMS2, and EPCAM, in these cancers (Figure 3B). CD73 expression was significantly linked to all MMR markers in KIRP, LGG, PAAD, and PRAD (*p* < 0.05). In addition, CD73 levels were positively related to tumor mutation burden (TMB) in BRCA, COAD, ESCA, GBM, OV, PAAD, SARC, SKCM, THYM, and UCEC, and negatively linked to TMB in BLCA, CESC, HNSC, LUAD, PCPG, PRAD and READ (Figure 3C; *p* < 0.05). Moreover, CD73 levels were positively linked to microsatellite instability (MSI) of COAD, READ, TGCT, and UCEC, while negatively linked to MSI of DLBC, GBM, LUAD, LUSC, PRAD, and SKCM (Figure 3D; *p* < 0.05).

### 3.5. Functional Analysis Based on CD73 Expression 

Enrichment analysis of pathways from the GSVA website revealed that CD73 participated in the activation of several immune-related pathways in most cancers, such as BLCA, BRCA, LUAD, LUSC, OV, and PRAD (Figure 4A). Interestingly, most of these pathways are associated with the activation and proliferation of T cells, macrophages, and fibroblasts (CAFs). These results are consistent with our previous data demonstrating the irreplaceable value of CD73 in regulating the immune activity of these cells. The most negatively enriched signaling pathways were focal adhesion, toll-like receptor signaling, and extracellular matrix (ECM) receptor interaction (Figure 4B; *p* < 0.0001), the most positively enriched signaling pathways were metabolism of xenobiotics by cytochrome P450 and phenylalanine metabolism from the KEGG mapping (Figure 4D; *p* < 0.05). The most negatively enriched signaling pathways were apoptosis, Kirsten rat sarcoma virus (KRAS) signaling up, and apical junction (Figure 4C; *p* < 0.0001), while the positively enriched signaling pathway was spermatogenesis from the HALLMARK mapping (Figure 4E; *p* < 0.05).

### 3.6. Single Cell Sequencing to Reveal CD73 Expression on Tumor and Stromal Cells

The co-expression of expression on tumor and stromal cells among several cancers was investigated based on the R package (copycat and infercnv) (Figure 5 and Appendix A). Cancer-associated fibroblasts (CAFs), hematopoietic progenitor cell-like (HPC), cancer cells, and thymic epithelial cells (TECs) were found to express high CD73 levels in HNSCC (Figure 5A), LIHC (Figure 5B), BRCA (Appendix A), CHOL (Appendix A) and OV (Appendix A). Meanwhile, cancer cells, T cells, B cells, M2 macrophages, CAFs, endothelial cells, and DCs were found to express CD73 in prostate cancer (Figure 5C), SKCM (Figure 5D), and STAD (Figure 5E). Moreover, CD73 was found to co-express on CAFs, neoplastic cells, B cells, macrophages, astrocytes, endothelial cells, and cancer cells in CHOL (Appendix A) and colorectal cancer (Appendix A).

### 3.7. Single Cell Sequencing to Analyze CD73 Expression and Related Signaling Pathways in GBM

Moreover, various cell types in GBM expressed elevated CD73, including oligodendrocyte progenitor cells, neurons, astrocytes, neoplastic, oligodendrocyte, neural stem cells, microglial cells, macrophages, M1 and M2 macrophages (Figure 6A). The cellular communication of CD73 expression and infiltrated cells was analyzed through the R package “CellChat”. These identified 14 cell types are grouped into four types based on their different roles in cellular communication: sender, receiver, mediator, and influencer. The cell patterns regarding the 14 cell types (receivers and senders) were classified into three different models (Appendix A). The specific genes associated with these receiver and sender communication patterns show three patterns (Appendix A). The dot plots (Appendix A) and river plots (Appendix A) depicted the communication patterns of the receiver and sender for 14 cell types. Astrocytes, neoplastic cells (high and low), and neural stem cells were correlated with the signaling pathways of receiver and sender in pattern 1. We further described the correlation between CD73 levels and corresponding signaling pathways. Neoplastic cells expressing CD73 exhibited strong interaction with macrophages through the CXCL, EGF, EPO, FASLG, FSH, GRN, MIF, MK, SPP1, TRAIL, VEGF, and VISFATIN signaling pathways (Figure 6B–M). Neoplastic cells expressing CD73 exhibited strong interaction with T cells through the CXCL, EGF, FASLG, MIF, MK, SPP1, TRAIL, and VISFATIN signaling pathways (Figure 6B–M).

### 3.8. Correlation between Macrophages, T Cells, and CD73 Expression

Previous results demonstrated that cancer cells and many stromal cell types in TME could express CD73, especially macrophages and T cells. Next, we explored the CD73 expression profile on macrophages and T cells using tissue chips of pan-cancer samples (Figure 7A–L). CD73 has been reported to be widely expressed in cancer cells, DCs, Tregs, NK cells, MDSCs, and TAMs. So, it is conceivable that different types of CD73-positive stained cells could be detected by immunofluorescence staining. To furthest resolve the impact of other cells, we tried to focus on the cancer cells with a relatively large nucleus. We chose CD68 as a macrophage marker, CD163 as an M2 macrophage marker, and CD8 as a T-cell marker (Figure 7M). Immunofluorescence staining showed that WHO Ⅲ gliomas had more CD73 expression than WHO Ⅱ gliomas (Figure 7A). Meanwhile, we also found that GBM shows higher CD73 levels than LGG and that CD73 was closely related to CD163-positive cells. In addition, the CD73 expression was elevated in tumor tissues than controls (paracancerous tissues) in LSCC and THCA. On the contrary, there were more CD73 levels in the controls than in tumor tissues in UTUC, BLCA, and TGCT. Furthermore, PRAD patients with higher Gleason scores had upregulated CD73 levels than patients with lower Gleason scores. We found that CESC, PSCC, and OV tissues expressed a large number of CD163-positive cells. These results are consistent with our above analysis using public databases and single-cell sequencing.

Moreover, these tumor tissues exhibited much more M2 macrophages and T cells than controls. By immunofluorescence staining, we also demonstrated that CD73 exhibits different expression characteristics (low and high levels) in the same type of tumor (Figure 8). This result may be used to explain the different prognoses of patients with the same tumor.

The correlation between CD73 and other markers of macrophages and T cells was also analyzed. The single-nucleotide variant (SNV) mutant rate of CD73 was 13% compared to CD163 (67%), CD8B (10%), CD8A (8%), MRC1 (8%), and CD68 (7%) (Appendix A). CD73 had a significant methylation difference between tumor and normal samples in BRCA, LUSC, and LIHC (Appendix A). The relationship between CD73 methylation and gene expression was significant in THCA, SKCM, HNSC, LIHC, and STAD (Appendix A). We then analyzed the signaling pathways related to these markers (Appendix A). EMT, RAS/MAPK, and apoptosis signaling were the most activated pathways associated with CD73 expression and also significantly linked to the expression of other markers. DNA damage response and hormone AR signaling were the top two inhibited pathways linked to the expression of CD73 and other markers. Next, we used the GDSC and CTRP websites to explore molecules that are sensitive to these markers (Appendix A). Based on CD73 expression, we predicted a lot of targeted small molecule compounds with promising therapeutic effects, providing a new direction for immunotherapy targeting the CD73 signaling pathways in these cancers.

### 3.9. Potential Therapeutic Values of CD73 in Immunotherapy Cohorts

Finally, the potential therapeutic effects of CD73 in these cancers were explored from the public datasets. Interestingly, in 25 immunotherapy cohorts, the AUC values for CD73-targeted therapy exceeded 0.5 in 11 cohorts (Figure 9A). The predictive role of CD73 was higher than TMB, T. Clonality, and B. Clonality. However, CD73 has a lower predictive effect than that of MSI score, CD274, TIDE, IFNG, and CD8, which had AUC values higher than 0.5 in 13, 21, 18, 17, and 18 immunotherapy cohorts, respectively. We also compared CD73 expression levels across different tumor models and ICB treatments, between pre- and post-ICB treatment and responders and non-responders. In addition, we found that CD73 significantly predicted response to immunotherapy in three mouse cohorts, with a significant increase in CD73 levels in sensitive responders (Figure 9B). Moreover, the COMPARE algorithm was applied to predict the potential anticancer compounds associated with the inhibition of CD73 signaling pathways. Interestingly, among the more than 90,000 pure molecules reported to have anticancer activity against NCI cancer cell lines, we identified only one synthetic compound, S820283, which specifically targets the CD73 signaling pathway. S820283 was reported to inhibit the growth of many cancerous cell lines, from the hematological, digestive, urinary, and central nervous systems, with a GI50 concentration (−log10) of −4.01 ± 0.06 lM, respectively (Figure 9C). We compared CD73 expression levels across cell lines between pre- and post-cytokine-treated samples. CD73 gene expression was compared across cell lines before and after cytokine treatment (Figure 9D). CD73 significantly predicted response to pre- and post-cytokine treatment in six murine immunotherapy cohorts.

## 4. Discussion

CD73 is vital in regulating adenosinergic signaling pathways under physiological and pathological conditions [43]. In cancer immunity, CD73 is a new immune checkpoint that promotes the formation of adenosine. Adenosine signaling is an essential component of inherent immune regulation. Adenosine and ATP are rarely expressed in extracellular fluids under normal conditions [44]. However, upregulation of extracellular ATP can be induced under pathological conditions, including inflammation, ischemia, or cancer. CD73 can progressively dephosphorylate extracellular ATP into adenosine, which eventually causes a large amount of adhesion outside the cell [45]. Extracellular adhesion exhibits a remarkable inhibiting effect on the tumor immune response, dampening effector cell function, and stabilizing immunosuppressive regulatory cells [46]. Blockage against the CD73/adenosine axis can effectively promote the anti-tumor progress of rectal tumor cells and improve the patient outcome with advanced rectal cancer without metastasis [47]. John Stagg et al. found that monoclonal antibodies targeting CD73 can mediate adaptive anti-tumor immunity and prevent breast cancer invasion [48]. Thus, in line with several classical immune checkpoints (PD-1/PD-L1 and CTLA-4), the blockage of the CD73-adenosine axis has been considered a grand promise for further improving clinical outcomes in cancer patients [49,50]. In this article, we systematically and comprehensively describe the role of CD73 in the immune process of multiple cancer tumors.

Overexpressed CD73 was associated with shorter OS and poor prognosis for patients in many cancers [51,52,53]. This paper detected the prognostic value of CD73 in pan-cancer and found that elevated CD73 was associated with poor OS in patients with ACC, BRCA, CESC, HNSC, KIRP, LGG, LIHC, LUAD, LUSC, MESO, STAD, UVM, KICH, PAAD, and TGCT. These data, together with previous studies, suggest an important role for CD73 in a variety of tumors. Increasing evidence demonstrated that multiple cell types express CD73 in the TME, such as tumor cells, stromal cells, infiltrated immune cells, and endothelial cells [54]. Multiple factors participate in the regulation of CD73 expression in the tumor microenvironment. For example, the study found that hypoxia-inducible factor-1 in the TME can upregulate the CD73, which finally help to protect the epithelial barrier under hypoxic condition [55]. In addition, other factors such as IL-6, IFN-1, TGF-β, TNF-α, Wnt, and STAT3 signaling also can stimulate the expression of CD73 [56,57].

In the current paper, we comprehensively clarified the expression aspect of CD73 in pan-cancer and various tumor cell lines. CD73 expression in tumor tissues significantly differed from in normal tissues among these cancers. Meanwhile, we revealed the mutational profiles of CD73 in these cancers. By single-cell sequencing and the Timer algorithm, we explored the correlation between CD73 expression and cancer cells, macrophages, B cells, M2 macrophages, neutrophils, T cells, Tumor Endothelial Cells (TECs), CAFs, and DCs in the tumor microenvironment. CD73 is very closely related to the activation, infiltration, and differentiation of these cells and other immune-related pathways. In particular, we focused on the network interaction of CD73 with immune cell signaling pathways in GBM. The cellular communication of CD73 expression and 14 cell types was investigated. These results are consistent with previous findings focusing on CD73 in tumor immunity.

Despite disappointing results in recent years, immunotherapy has emerged as a promising windfall for many cancers [58,59]. Tumor immunotherapies include strategies such as tumor vaccines, oncolytic viruses, and antibodies blocking immune checkpoint pathways [60]. In particular, the CTLA-4 and PD-1 mechanisms are the top two representative immune checkpoint pathways, which negatively mediate the immune characteristic of T cells during tumor immunity [11,61]. Therefore, immune checkpoint inhibitors alone or in combination with traditional therapies, including surgery, chemotherapy, or radiotherapy, are expected to become the standard strategy for the first-line treatment of cancers in the next years [62]. In this paper, overexpressed CD73 was found to be significantly associated with neoantigen levels in KIRC, HNSC, and CESC. Meanwhile, the relationship between CD73 and MR markers, TMB, and MSI of these tumors was also elaborated. In addition, high levels of CD73 showed an intimate relationship with several classical immune checkpoints, such as NRP1, CD276, PD-L2, PD-L1, and CD44. Enrichment analysis of pathways revealed that elevated CD73 plays a vital role in mediating the immune pathways associated with the activation of T cells, macrophages, and CAFs. These observations emphasize the complex and vital immunosuppressive role of CD73 in the TME. More importantly, it provides a new direction for immunotherapy against CD73 as it is a new immune checkpoint for many cancers. Moreover, we found good potential therapeutic value of CD73 in eleven immunotherapy cohorts and three mouse cohorts; and we found that CD73 significantly predicted the response to pre- and post-cytokines in six cell line immunotherapy cohorts. These data will likely provide partial theoretical support for future clinical trials targeting CD73 in tumor immunotherapy.

TAMs are the primary component of stromal cells that can promote the formation of an immunosuppressive environment by producing various cytokines and chemokines and activating the inhibitory immune checkpoint molecules from other immune cells [63]. Activated macrophages are usually polarized to M1 and M2 subtypes under different stimuli. These two macrophages have different surface markers, metabolic features, and genetic backgrounds. Generally, M1 macrophages belong to classical-activated phenotypes that boost inflammation response against pathogens and tumor cells. On the contrary, M2 macrophages are alternative-activated phenotypes that exhibit an immunosuppressive property that promotes tissue recovery and tumor progression [64]. M1 macrophages secrete pro-inflammatory cytokines such as TNFαand IL-1β; their markers include CD16, CD86, MHC2, and iNOS. M2 macrophage produces anti-inflammatory cytokines such as IL-10 and TGFβ; their markers include arginase-1, CD206, Ym-1, and CD36 [65]. Previous studies showed that adenosine could activate macrophages and regulate their phagocytic function via A_2A_, A_2B_, and A_3_ receptors [66,67]. Interestingly, this article found that CD73 is closely related to macrophages and M2 macrophages, and we validated the association between CD73 and these two cellular markers using public databases. The correlation between CD73 and CD68 and CD163 was also observed using multiplex immunofluorescence. Based on the above results, we speculate that CD73 may affect the immune properties of macrophages in the TME by producing multiple adenosines to promote tumor progression.

T cells and cancer-associated fibroblasts are two critical parts full of heterogeneity and plasticity in TME. These cells were also found to express CD73 in previous studies. With special antigen-directed cytotoxicity, T lymphocyte has become a powerful tool in cancer immunotherapy in recent years [68]. For example, foxp3-expressing regulatory T cells (Treg) can inhibit anti-tumor immunity by releasing immune checkpoint proteins such as CD25 and CTLA-4 and are often associated with unfavorable outcomes in cancer patients [69]. Therefore, strategies focusing on the depletion of Treg cells have proven effective in activating anti-tumor immune response [70]. In this study, we likewise found an upregulation of CD73 expression in certain tumors on T cells. Immunofluorescence staining and public site prediction analysis also confirmed that CD8 levels are closely associated with CD73 expression. CAFs are a group of activated fibroblasts that can regulate oncogenesis, tumor invasion, and immune resistance [71]. Several mechanisms can induce CAFs activation in TME under pathological conditions, including IL-1/NF-κB and IL-6/STAT signaling [72]. However, the specific mechanisms of how CD73 is involved in modulating their cell functions remain unclear and are needed to be elucidated in the future.

Therapeutic resistance has become the principal limiting factor for patients’ complete response to cancer therapies during the past decades. Before or after treatment, drug resistance remains one of the leading causes of death in patients with cancer [73]. Several mechanisms lead to drug resistance, including drug inactivation and efflux, drug target alteration, cell death inhibition, DNA damage repair, and tumor heterogeneity [74]. Therefore, reducing drug resistance or finding sensitive drugs is critical in tumor therapies. In this present paper, we recognized the potent immunotherapy role of CD73 in multiple cancers. Moreover, we explored a series of targeted and small molecule drugs with promising therapeutic effects based on the expression of CD73, which might provide novel strategies for treatment against CD73 in pan-cancer. We identified S820283 as a potential anticancer compound associated with inhibition of the CD73 signaling pathway. However, there are still many limitations in our research. First of all, we detected widespread expression of CD73 on tumor cells and immune cells in the tumor microenvironment, but the effect of CD73 on the biological behavior of these cells has not been explored in depth. In particular, the specific role of CD73-positive macrophages as well as Tregs in tumor immunity has not been experimentally verified. Second, the data used in this paper are selected from public databases for analysis and lack our own clinical data. We will collect clinical data from glioma patients for validation in our subsequent work. In addition, prognostic data on tumor immunotherapy targeting high CD73 expression in the tumor microenvironment should also be collected. Third, there is a lack of treatment consistency when studying pan-cancer analysis, particularly when utilizing CD73 as a biomarker for OS. These limitations should be expanded to explore future research efforts by other groups. Last but not least, the necessary cellular experiments or animal experiments to verify the effects on the tumor immune microenvironment when CD73 expression is knocked down or up-regulated are lacking in this paper, and further studies need to be enhanced in the future.

## 5. Conclusions

We systematically discussed the prognostic value and immune aspects of CD73 in pan-cancer. Therapies targeting CD73 in the TME may become a useful tool for pan-cancer immunotherapy.

## Figures and Tables

**Figure 1 cancers-14-05663-f001:**
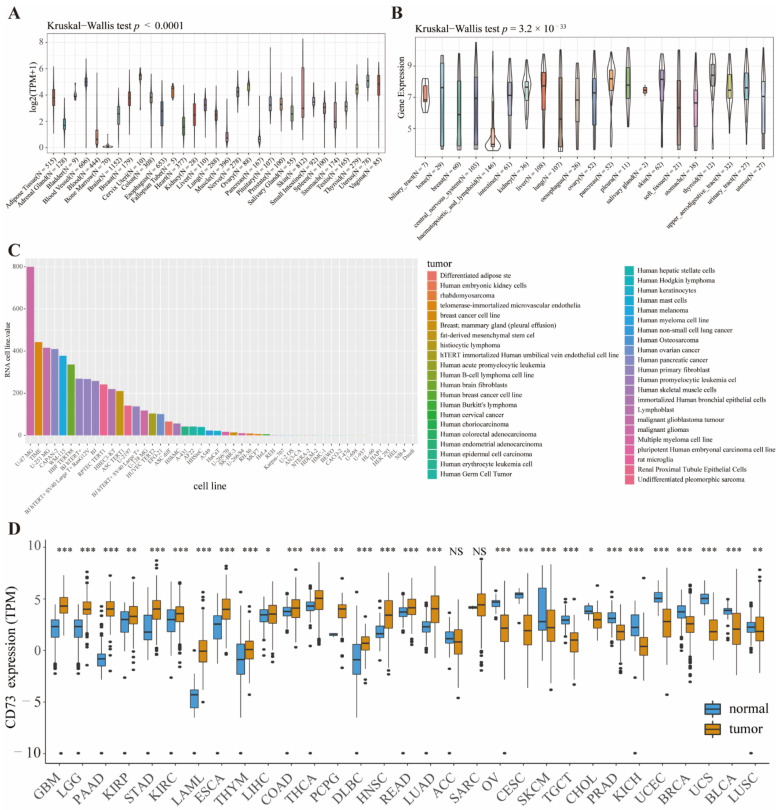
CD73 is widely expressed in cancer cell lines, normal and tumor tissues. CD73 expression in human normal tissues (**A**). CD73 expression in cancer cell lines from CCLE (**B**) and HPA (**C**) databases. CD73 expression in normal and tumor tissues from TCGA and GETx (**D**) databases. * *p*< 0.05, ** *p* < 0.01, *** *p* < 0.001, NS: no significant differences.

**Figure 2 cancers-14-05663-f002:**
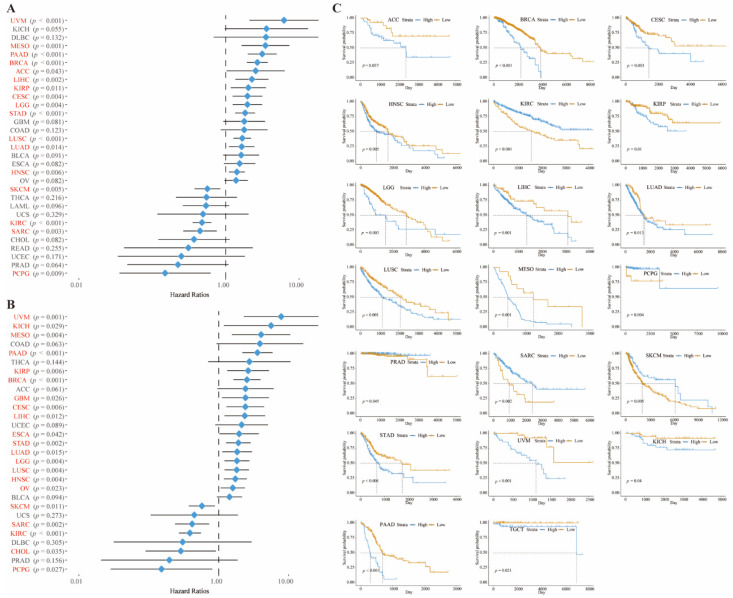
CD73 exhibits significant prognostic value in a variety of cancers. Forest plot showed the prognostic role of CD73 in OS (**A**) and DSS (**B**) analysis. KM method showed the prognostic role of CD73 in OS analysis (**C**).

**Figure 3 cancers-14-05663-f003:**
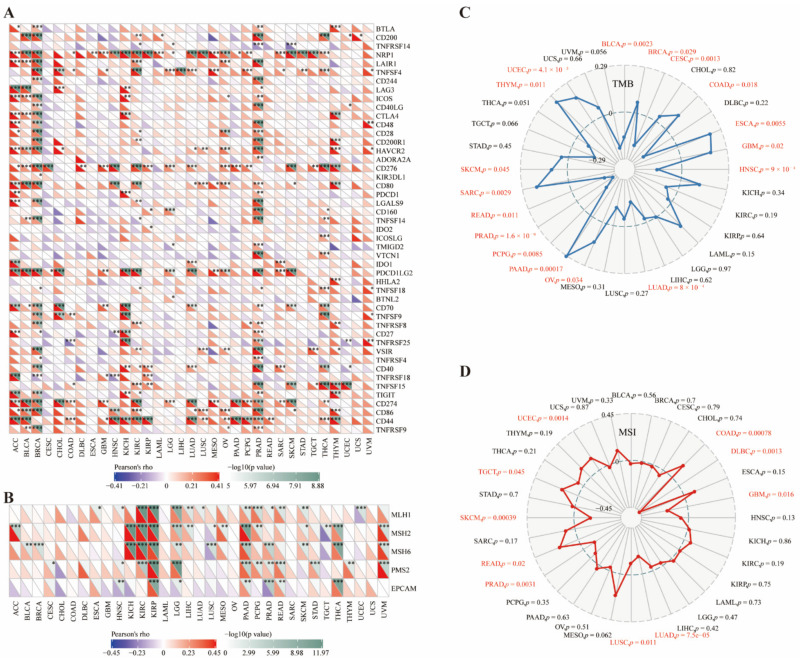
CD73 shows close correlation with immune checkpoints, DNA MMR markers, TMB, and MSI. Relationship between CD73 levels and classical immune checkpoints (**A**), DNA MMR markers (**B**), TMB (**C**), and MSI (**D**). * *p*< 0.05, ** *p* < 0.01, *** *p* < 0.001.

**Figure 4 cancers-14-05663-f004:**
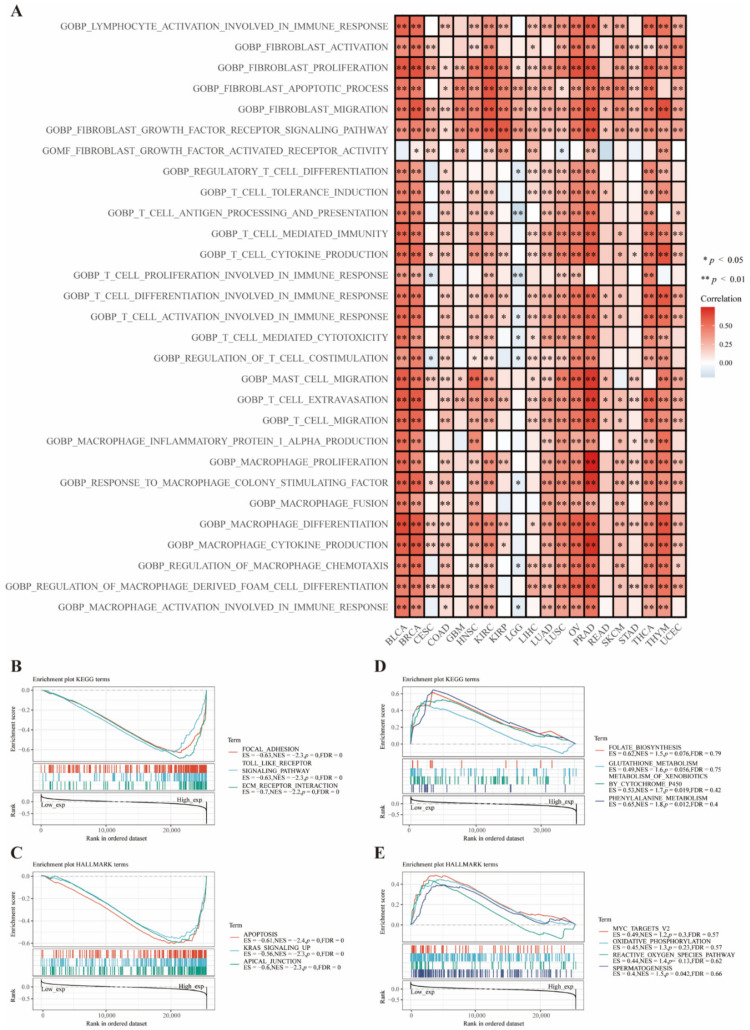
Functional enrichment pathways analysis of CD73. Enriched signaling pathways of CD73 from the GSVA algorithm (**A**). Top three negative (**B**) and top four positive (**D**) enriched KEGG terms of CD73. Top three negative (**C**) and top four positive (**E**) enriched HALLMARK terms of CD73.

**Figure 5 cancers-14-05663-f005:**
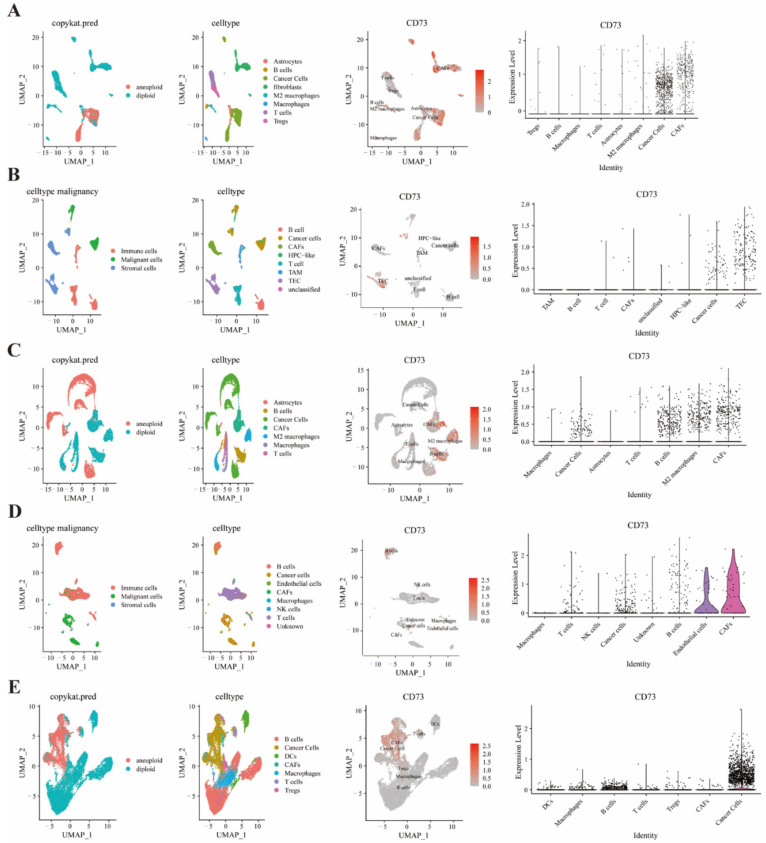
Single-cell analysis shows that CD73 is significantly expressed on tumors and some immune cells. CD73 expression on various cell types in TME based on the R package in HNSCC (**A**), LIHC (**B**), prostate cancer (**C**), SKCM (**D**), and STAD (**E**).

**Figure 6 cancers-14-05663-f006:**
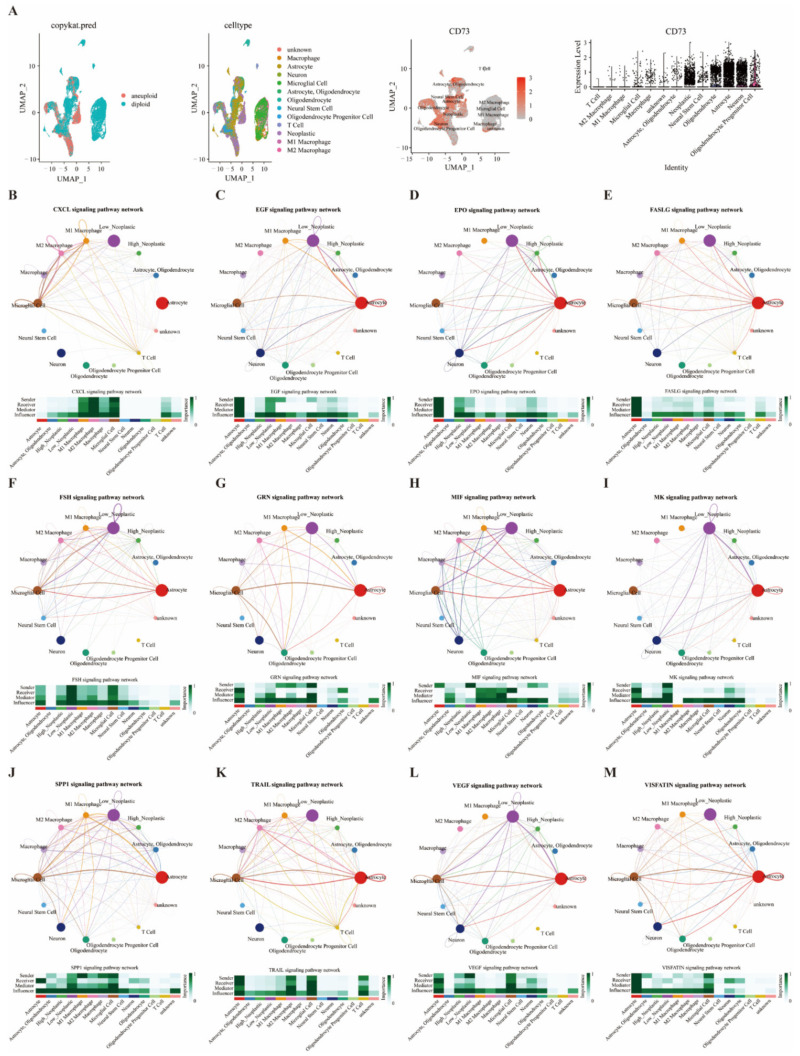
Single cell analysis shows that CD73 is closely related to immune cells and immune pathways in GBM. CD73 expression in microenvironment of GBM (**A**). Cross network diagram of signaling pathways between tumor cells and infiltrated cells based on CD73 expression (**B**–**M**).

**Figure 7 cancers-14-05663-f007:**
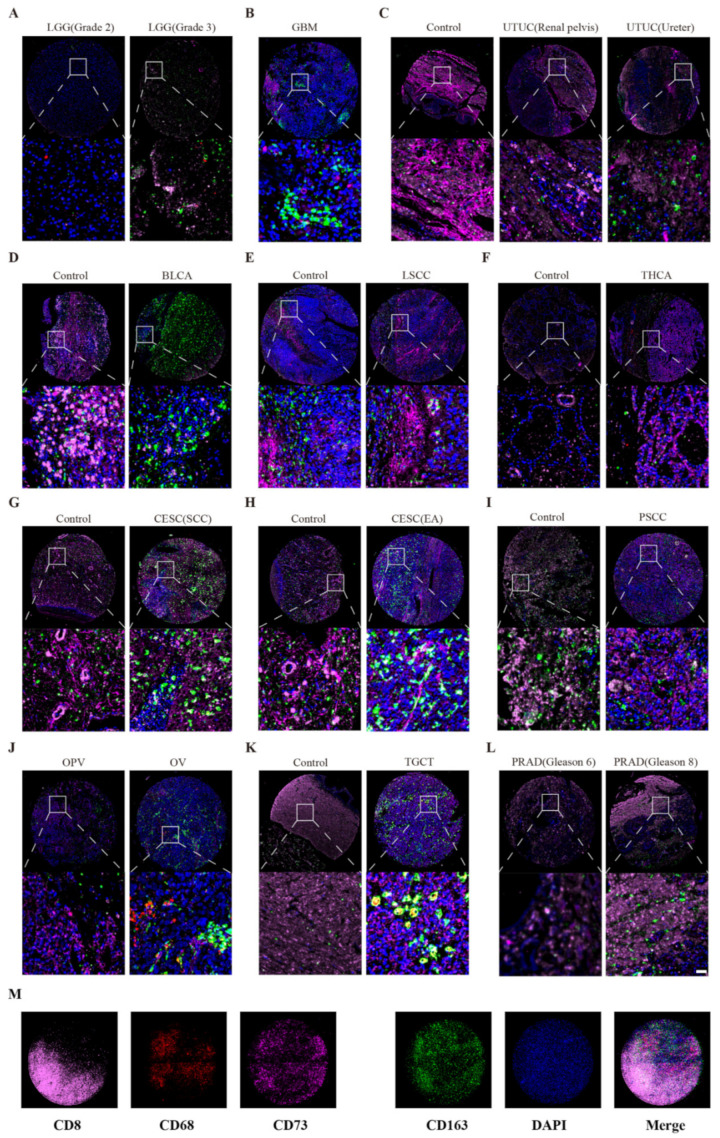
Triple immunofluorescences analyze the co-expression of CD73 in pan-cancer chip. Relationship between CD73 and CD8, CD163 and CD68 expression in LGG (**A**), GBM (**B**), UTUC (**C**), BLCA (**D**), LSCC (**E**), THCA (**F**), CESC (**G**, **H**), PSCC (**I**), OV (**J**), TGCT (**K**), and PRAD (**L**), Immunofluorescence markers (**M**). Scale bar = 40 μm.

**Figure 8 cancers-14-05663-f008:**
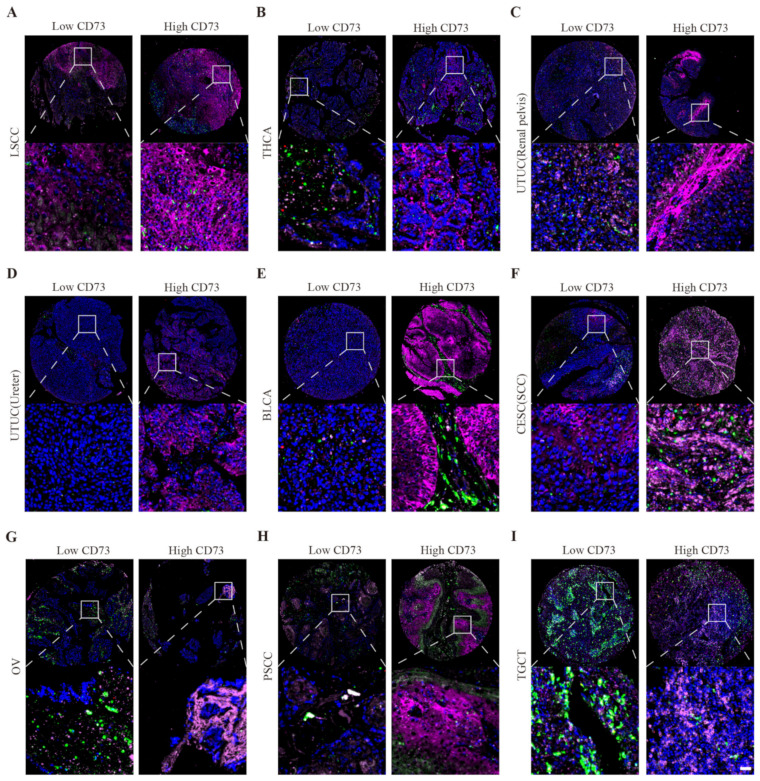
Triple immunofluorescences show high and low expression of CD73. Low and high CD73 expression in LSCC (**A**), THCA (**B**), UTUC (**C**,**D**), BLCA (**E**), CESC (**F**), OV (**G**), PSCC (**H**), and TGCT (**I**). Scale bar = 40 μm.

**Figure 9 cancers-14-05663-f009:**
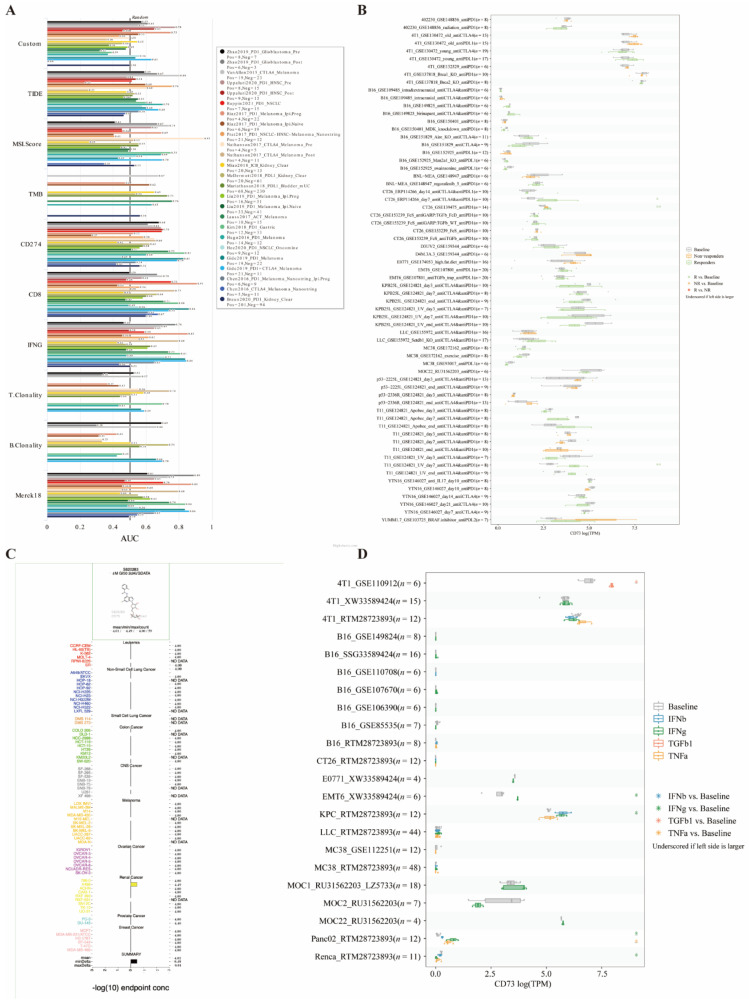
CD73 has high therapeutic values in the immunotherapy cohort. Immunotherapy response in cohorts (**A**) and in vitro (**B**), potent anticancer inhibitors (**C**), and Immunotherapy response in vivo (**D**) of CD73 were identified from the public databases.

## Data Availability

Data used in this work can be acquired from the GEO, the TCGA, and the GETX database and so on. Further original data inquiries can be directed to the corresponding authors.

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
