# Peer review of "Identification of CD73 as a Novel Biomarker Encompassing the Tumor Microenvironment, Prognosis, and Therapeutic Responses in Various Cancers"

_cancers, 2022, doi:10.3390/cancers14225663_

Round 1

Reviewer 1 Report

In their manuscript, Tang and Zhang et al. used a comprehensive data analysis approach to explore the prognostic value and immune aspects of CD73 in severe cohorts of 33 multiple cancers. The work was further validated by imaging in cancer tissues. The findings would be of general interest to the community of cancer researchers.Overall, the authors have highlighted a vital topic; however, a few concerns need to be addressed.

1-     Fig. 2C is blurry and missing clarity

2-     In Fig.5 A-E and Fig.6A, the color scheme seems unnecessary for the plots showing expression levels of CD73 in different TME cells.

3-     The discussion is wordy and resembles introduction, the authors should provide a more specific discussion of the findings. 

Author Response

Comments:

# Reviewer 1:

Comments and Suggestions for Authors

In their manuscript, Tang and Zhang et al. used a comprehensive data analysis approach to explore the prognostic value and immune aspects of CD73 in severe cohorts of 33 multiple cancers. The work was further validated by imaging in cancer tissues. The findings would be of general interest to the community of cancer researchers. Overall, the authors have highlighted a vital topic; however, a few concerns need to be addressed.

Response: Thank you for your review and recognition of our paper. Your comments are very important to improve the quality of this paper.

1、2C is blurry and missing clarity

Response: Thank you for your valuable advice. We have enlarged the size of the text on the Fig. 2C and improved the clarity of this image. Please contact us if there are any further questions.

2、In Fig.5 A-E and Fig.6A, the color scheme seems unnecessary for the plots showing expression levels of CD73 in different TME cells.

Response: Thank you for your valuable advice. We have removed the color scheme from Fig.5 A-E and Fig.6A. Please contact us if there are any further questions.

3、The discussion is wordy and resembles introduction, the authors should provide a more specific discussion of the findings.

Response: Thank you for your valuable advice. We reorganized the discussion section with a specific discussion of the findings. Please see the “Discussion” section for details.

Reviewer 2 Report

Tang et al propose that CD73 is a novel biomarker that has been determined by several studies prior to their own investigation, with little knowledge of the mechanistic or pathways associated with, nor the effects or interactions within the TME of this CD molecule. 

Their goals were to use publically available databases, alongside tissue microarrays and single-cell sequencing to better elucidate CD73 interactions and why it might be a plausible target for anti-cancer therapy. A specific hypothesis is not stated and should be revised in the text. 

I commend the authors on the amount of work pursued for this manuscript and it is both highly relevant and well-executed. Appropriate tissue controls have been used for expression analysis. 

Suggestions:

1. The methods are brief and have many grammatical and language/scientific language errors, please revise.

2. Line 126 - annotation of non-tumor cells based on 'specific markers' - I would ask that specific markers are listed. 

3. Please ensure all databases are appropriately referenced.

4. In some instances the "top three" are listed and in others the "top 5" are listed. I would suggest revision to be both consistent and biologically relevant to the author's main goals and discussion points. 

5. Several figures are of poor resolution, making some interpretations difficult, please revise for the final publication.

6. Figure legends should be revised with explanation/interpretation or statistical inference. These should be standalone figures that are self-explanatory; particularly the single-cell analysis figures (Figures 5 and 6). 

7. Line 320-321 - Reference is missing

8. Figure 7 - there are arrows present and it is unclear to the reader what these are highlighting. Please revise

9. Limitations of the current study, other than Tregs not being evaluated here, were listed in a single and very brief sentence (465-466). Please revise. One limitation that should be touched upon is the lack of treatment consistency when looking at pan-cancer analysis, especially when utilizing CD73 as a biomarker for OS. These limitations should be expanded to probe future research endeavors by other groups. 

Author Response

Reviewer 2

Tang et al propose that CD73 is a novel biomarker that has been determined by several studies prior to their own investigation, with little knowledge of the mechanistic or pathways associated with, nor the effects or interactions within the TME of this CD molecule. 

Their goals were to use publically available databases, alongside tissue microarrays and single-cell sequencing to better elucidate CD73 interactions and why it might be a plausible target for anti-cancer therapy. A specific hypothesis is not stated and should be revised in the text. I commend the authors on the amount of work pursued for this manuscript and it is both highly relevant and well-executed. Appropriate tissue controls have been used for expression analysis. 

Response: Thank you for your comments and compliments on our paper. Your comments are very important to improve the quality of this paper. We presented our hypothesis in the "Introduction" section and added the sentence, "Therefore, we propose that CD73 may affect tumorigenesis and progression by influencing the biological functions of immune cells in the tumor microenvironment." The discussion was also reorganized and our hypothesis was tested with the results of this paper. Please see the details in our paper.

Suggestions:

1、The methods are brief and have many grammatical and language/scientific language errors, please revise.

Response: Thank you for your valuable advice. We have described the experimental methodology in detail and revised the syntax and linguistic/scientific language errors. Please see the revision details in the "Materials and Methods" section.

2、Line 126 - annotation of non-tumor cells based on 'specific markers' - I would ask that specific markers are listed. 

Response: Thank you for your valuable advice. We have listed the specific markers in the “Materials and Methods” section. We have added this term “The specific markers of non-tumor cells are listed as following: fibroblast (COL3A1), M2 macrophage (CD163, CD68, MRC1), B cell (CD79A), plasma cell (JCHAIN), T cell (CD3D, CD8A, CD4), NKs (GNLY, NKG7, EOMES, KIR2DL3, GZMA), astrocyte (ALDH1L1, SLC1A3, SLC1A2, GFAP), immune cell (PTPRC), Tregs (FOXP3, IL2RA), DC (CD80, CD86, CD40), FDCs (CR2, FCER2, CR1), plasma cell (SDC1, TNFRSF17), neutrophils (CEACAM8), ILC1 (TBX21, IKZF3, CXCR3), ILC2 (GATA3, MAF, PTGDR2, HPGDS), ILC3 (RORC, IL23R, IL1R1, KIT).” in the “2.3 Single-cell sequencing analysis” section.

3、Please ensure all databases are appropriately referenced.

Response: Thank you for your warm advice. We have cited all databases appropriately. Please contact us if there are any further questions.

4、In some instances the "top three" are listed and in others the "top 5" are listed. I would suggest revision to be both consistent and biologically relevant to the author's main goals and discussion points. 

Response: Thank you for your valuable advice. Because CD73 is widely expressed in normal and cancer and its association with immune infiltration in multiple tumors are very close. Therefore we have listed only the first three most prominent tumor types. We have revised it and made it consistent with our main objectives and discussion points.

5、Several figures are of poor resolution, making some interpretations difficult, please revise for the final publication.

Response: Thank you for your warm advice. We have improved the resolution of images such as Figure 1, Figure 2, Figure 3, Figure 5, and Figure S2. Please contact us if there are any further questions.

6、Figure legends should be revised with explanation/interpretation or statistical inference. These should be standalone figures that are self-explanatory; particularly the single-cell analysis figures (Figures 5 and 6). 

Response: Thank you for your valuable advice. We have modified our figure legends accordingly. Please see the details in the figure legends. Please contact us if there are any further questions.

7、Line 320-321 - Reference is missing

Response: Thank you for your valuable advice. What we mean is that the multiplex fluorescence results of Figure 7 match some of our previous results analyzed with public databases and single-cell sequencing. We have modified the statement in order to avoid misunderstandings. We changed the original sentence to this term“These results are consistent with our above analysis using public databases and single-cell sequencing”. Please contact us if there are any further questions.

8、Figure 7 - there are arrows present and it is unclear to the reader what these are highlighting. Please revise

Response: Thank you for your kind advice. We have removed these arrows. Please contact us if there are any further questions.

9、Limitations of the current study, other than Tregs not being evaluated here, were listed in a single and very brief sentence (465-466). Please revise. One limitation that should be touched upon is the lack of treatment consistency when looking at pan-cancer analysis, especially when utilizing CD73 as a biomarker for OS. These limitations should be expanded to probe future research endeavors by other groups. 

Response: Thank you for your kind advice. We have added these limitations to our paper. We have added the term “However, there are still many limitations in our research. First of all, we detected widespread expression of CD73 on tumor cells and immune cells in the tumor microenvironment, but the effect of CD73 on the biological behavior of these cells has not been explored in depth. In particular, the specific role of CD73-positive macrophages as well as Tregs in tumor immunity has not been experimentally verified. Second, the data used in this paper are selected from public databases for analysis and lack our own clinical data. We will collect clinical data from glioma patients for validation in our subsequent work. lack of cohort validation. In addition, prognostic data on tumor immunotherapy targeting high CD73 expression in the tumor microenvironment should also be collected. Third, there is a lack of treatment consistency when studying pan-cancer analysis, particularly when utilizing CD73 as a biomarker for OS. These limitations should be expanded to explore future research efforts by other groups. Last but not least, the necessary cellular experiments or animal experiments to verify the effects on the tumor immune microenvironment when CD73 expression is knocked down or up-regulated are lacking in this paper, and further studies need to be enhanced in the future.” in the “Discussion” section.
